# Wood Anatomical and Demographic Similarities Between Self-Standing Liana and Tree Seedlings in Tropical Dry Forests of Colombia

**DOI:** 10.3390/plants13213023

**Published:** 2024-10-29

**Authors:** Juliana Puentes-Marín, Andrés González-Melo, Beatriz Salgado-Negret, Roy González-M, Julio Abad Ferrer, Juan Pablo Benavides, Juan Manuel Cely, Álvaro Idárraga-Piedrahita, Esteban Moreno, Camila Pizano, Nancy Pulido, Katherine Rivera, Felipe Rojas-Bautista, Juan Felipe Solorzano, María Natalia Umaña

**Affiliations:** 1Instituto de Biología, Universidad de Antioquia, Medellin 050010, Colombia; juliana.puentes@udea.edu.co; 2Department of Ecology and Evolutionary Biology, University of Michigan, Ann Arbor, MI 48109, USA; maumana@umich.edu; 3Departamento de Biología, Universidad Nacional, Bogota 111321, Colombia; 4Departamento de Ciencias Forestales, Universidad del Tolima, Ibague 110931, Colombia; 5Dirección Territorial Caribe, Parques Nacionales Naturales de Colombia, Santa Marta 110221, Colombia; 6Fundación Jardín Botánico de Medellín, Herbario “Joaquín Antonio Uribe” (JAUM), Medellin 050010, Colombia; 7Facultad del Medio Ambiente y Recursos Naturales, Universidad Distrital Francisco José de Caldas, Bogota 110711, Colombia; 8Department of Biology, Lake Forest College, Lake Forest, IL 60045, USA

**Keywords:** growth, wood anatomy, seedlings, lianas, trees, tropical dry forests

## Abstract

Canopy lianas differ considerably from trees in terms of wood anatomical structure, and they are suggested to have a demographic advantage—faster growth and higher survival—than trees. However, it remains unclear whether these anatomical and demographic differences persist at the seedling stage, when most liana species are self-standing and, consequently, might be ecologically similar to trees. We assessed how self-standing liana and tree seedlings differ in relation to wood anatomy, growth, and survival. We measured 12 wood traits and monitored seedling growth and survival over one year for 10 self-supporting liana and 10 tree seedling species from three tropical dry forests in Colombia. Liana and tree seedlings exhibited similar survival rates and wood anatomies for traits related to water storage and mechanical support. Yet, for traits associated with water transport, liana seedlings showed greater variability in vessel lumen size, while tree seedlings had higher vessel density. Also, the liana relative growth rate was significantly higher than for trees. These results indicate that, while self-supporting liana and tree seedlings are anatomically similar in terms of mechanical support and water storage—likely contributing to their similar survival rates—liana seedlings have a growth advantage, possibly due to more efficient water transport. These findings suggest that the well-documented anatomical and demographic differences between adult lianas and trees may depend on the liana’s developmental stage, with more efficient water transport emerging as a key trait from early stages.

## 1. Introduction

Lianas are a key component of lowland tropical forests, as they play an important role in forest dynamics, functioning, and management [1,2,3,4,5]. Over the last decades, liana abundance relative to trees has increased in many tropical forests [5,6], which has been suggested to have detrimental effects on tree demography and forest functioning, as lianas can limit tree recruitment, growth, and survival, thus decreasing forest capacity to sequester and store CO_2_ [2,7,8,9,10]. Understanding life-history variations between lianas and trees is, therefore, necessary for elucidating the factors mediating the interactions of these two life forms, and the mechanisms driving liana relative abundance [11]. Canopy trees and lianas differ substantially in both their stem allocation patterns, demographic rates [11,12,13,14,15]. For instance, unlike self-supporting trees, lianas rely on other plants to get access to the canopy, and consequently can prioritize stem water transport and storage over mechanical support [12,13]. This stem allocation pattern appears to be particularly advantageous in drier environments, such as seasonal dry forests, where lianas tend to grow better than co-occurring trees [14,16]. Evidence suggests, however, that these stem allocation and growth differences between lianas and trees are probably age- or size-dependent [17]. In terms of liana survival, previous studies have suggested that canopy lianas may be more susceptible to drought-related mortality than trees, as they tend to exhibit hydraulically efficient traits that make them more vulnerable to cavitation during drought conditions [9,18,19], although this contradicts studies showing that canopy lianas are increasing in abundance [5,20]. At the seedling stage, most liana species are self-standing (e.g., [21]), which may imply more morphological and functional similarities between tree and liana seedlings than between these two groups in adult stage and potentially similarities in demography as well [1,21]. We are aware of no studies that have compared stem anatomical traits between liana and tree seedlings, and studies comparing liana and tree seedling demography have been scarce and show no conclusive results (e.g., [8,17,22,23]). The seedling stage represents a key demographic bottleneck determining species abundances and composition for advanced life stages [24,25]. Thus, comparing stem anatomical traits, survival, and growth between liana and tree seedlings is therefore an important step in gaining mechanistic insight into observed and potential shifts in abundance patterns impacting forest functioning.

It is well known that canopy trees and lianas have different wood anatomical structures (e.g., [12,13,26]). For instance, one of the most distinctive wood anatomical features of lianas is their vascular system [12,26,27]. A long-standing notion in wood anatomy is that lianas have exceptionally wider vessels compared to trees ([27]; and references therein). However, when accounting for stem length, given that plant size is a main driver of vessel lumen size [28], liana vessels are only marginally wider than tree vessels [27]. Despite this, adult lianas do have a distinctively wider variance in vessel lumen diameter and a higher vessel density (i.e., number of vessels per unit area; [26,27,29]). Together, these vessel traits contribute to the high hydraulic efficiency of lianas and likely allow them to avoid embolisms [29,30], which can explain, in part, their higher relative abundance in seasonal dry environments [16]. Beyond these vascular differences, canopy lianas and trees also diverge in terms of stem allocation to mechanical support and storage. Given that lianas have lower mechanical requirements than trees, they typically have lighter wood and allocate a lower stem volume to fibers, or have thinner fiber walls, in comparison to trees (e.g., [12]). This, in turn, allows lianas to allocate a larger stem volume to parenchyma cells, in particular to axial ones, which can favor stem flexibility and wounding recovery, storage capacity, and hydraulic safety [12,13,31]. Whereas these vascular, biomechanical, and storage-related differences are well established for canopy-level lianas and trees [12,13], it is unclear if they hold true at the seedling stage. For instance, most liana species are self-supporting as seedlings [1,21], suggesting that liana and tree seedlings would have similar biomechanical-related traits such as wood density, fiber fraction, or fiber wall thickness. Furthermore, the important role of parenchyma cells in favoring liana stem flexibility and healing wounds that result from stem twisting and bending [12,32], might only be relevant during the liana climbing phase, implying similar parenchyma fractions between trees and lianas at the seedling stage.

Canopy lianas and trees may differ not only anatomically, but also in their demographic strategies (e.g., [11,16,23,33]). Lianas are assumed to grow faster than trees (e.g., [11,16]), in part because lianas have less stem mechanical requirements and consequently can allocate more resources to growth [34]. Indirect evidence that lianas would grow more than trees comes from physiological studies showing higher photosynthesis and water potentials in lianas than in trees [11]. Lianas can also recruit better and exhibit a survival advantage than trees at the seedlings stage [23,33,34]. Although studies that have compared the demography between lianas and trees are still limited, they seem to support the notion that lianas have a demographic advantage over trees [11,14]. However, for seedlings trends are still controversial. Pasquini et al. [8], for instance, found that liana seedlings grew more than tree seedlings during a short time period; while for longer time periods, Gilbert et al. [17] and Umaña et al. [22] reported no growth differences between tree and liana seedlings. Additionally, Umaña et al. [23] found that liana proliferation is likely related to a survival advantage that emerges in early stages and is influenced by climatic conditions and past disturbance. This suggests that more research is needed to clarify the growth differences between lianas and trees at the seedling stage.

In this study, we examined wood anatomical and demographic differences between common self-standing liana and tree seedling species from three tropical dry forests. Our aim was to answer the following questions: (i) To what extent 0073elf-supporting liana and tree seedlings differ in terms of vascular, biomechanical, and storage-related wood anatomical traits? (ii) Do self-standing liana seedlings have an advantage compared to tree seedlings in terms of growth and survival? We hypothesize that liana and tree seedlings will have similar biomechanical (i.e., wood density, fiber fractions, and fiber wall thickness) and storage-related (i.e., axial and radial parenchyma fractions) traits. Yet, we expect liana seedlings to have higher variance in the vessel diameter and vessel density in comparison to tree seedlings. We also anticipate that liana seedlings will have a demographic advantage compared to tree seedlings, showing greater growth and higher survival rates than trees.

## 2. Results

The first and second PCA axes explained 20.09% and 31.06% of total interspecific variation, respectively. The first PCA axis was defined positively by VarDh, Dh, and negatively by Vn and Vd; while the second PCA axis was positively related with Ff and VarDh and negatively with Tpf and Apf. The PCA indicated that there was not a clear functional differentiation between liana and tree species as points were not spatially segregated (Figure 1).

### 2.1. Wood Anatomical Differences Between Liana and Tree Seedlings

Of the eight vessel traits measured, liana and tree seedlings differed only in two (Table 1). Liana seedlings showed a greater variation in hydraulically weighted vessel diameter (*t* = 2.12, *p* = 0.048), while tree seedlings had a higher vessel density (*t* = −2.39, *p* = 0.028; Table 1, Appendix A). Moreover, both growth forms did not differ in terms of wood density, fiber traits, or parenchyma fractions (Table 1).

#### Differences in Relative Growth Rate (RGR) and Survival Between Liana and Tree Seedlings

We found that growth form (i.e., liana or tree) had a significant effect on seedling relative growth rate (Figure 2, Table 2). Specifically, liana seedlings grew faster than tree seedlings (X^2^ = 8.26, *p* < 0.001, R^2^ conditional = 0.16, R^2^ marginal = 0.15). In contrast, we found that liana and tree seedlings did not vary in relation to survival (X^2^ = 0.49, *p* > 0.051, R^2^ conditional = 0.69, R^2^ marginal = 0.006). 

## 3. Discussion

In this study, we compared wood anatomical traits, as well as growth and survival rates, between 10 self-standing liana and 10 tree seedlings from three tropical dry forests. We found that, except for variance in vessel lumen and vessel density, liana and tree seedlings had similar wood anatomical structures, and did not differ in terms of their survival. Yet, liana seedlings grew faster than tree seedlings. This suggests that, at the seedling stage, lianas and trees may exhibit similar functionality in relation to their mechanical support and water storage. However, differences in water transport may give lianas a growth advantage. From these results we infer that the known anatomical differences between lianas and trees may depend, among other factors, on the developmental stage of the lianas. Below, we discuss in more detail these results and their implications for understanding the interaction between lianas and trees at the seedling stage.

### 3.1. Wood Anatomical Differences and Similarities Between Liana and Tree Seedlings

We predicted that self-standing liana and tree seedlings would not differ from tree seedlings regarding their wood anatomy related to mechanical support and storage, but that they would differ in their vascular anatomical characteristics. Our results support this prediction, demonstrating no significant differences in biomechanical and storage-related wood anatomy traits between the two groups. However, we did find that liana seedlings had lower vessel (i.e., fewer vessels per unit area), which agrees with some studies on adults (i.e., [26]), although this vessel trait difference is not consistently observed across all studies (see [12]). Moreover, we found that self-standing liana seedlings had a higher variance in vessel lumen size. This is one of the most distinctive anatomical features of lianas [26,29], which is thought to represent a hydraulic advantage given that wide vessels maximize water transport when water is not limiting, while narrow ones, which tend to be less vulnerable to drought-induced embolisms [30], can maintain water transport during drought stress [26,29].

Despite the trait differences mentioned above, many of the traits we studied did not show significant differences between liana and tree seedlings. In particular, vessel fraction or vessel lumen size did not differ between the two groups, which suggests that the higher vessel density of tree seedlings may not necessarily translate into higher xylem water conductivity [35]. A common assumption in comparative wood anatomy is that lianas have vessels considerably wider in comparison to trees ([13,27]; and references therein). While it is true that, for a given stem diameter, lianas have wider vessels than trees [28], stem diameter may not always be a good basis of comparison because vessel lumen size is better predicted by stem length than diameter [28]. In fact, some recent studies have shown that when accounting for stem length, lianas have vessels similar to, or just slightly wider than, trees [13,27]. Our finding that liana and tree seedlings had similar average vessel lumen size adds to these recent studies and may be partially explained by the fact that we compared liana and tree seedling of similar heights, and measured wood anatomical traits at the same stem height (i.e., at 5 cm from the base). This highlights the importance of considering plant height when comparing vessel traits across individuals or species [13,28].

The fact that adult lianas lean on trees for support is a key factor mediating wood anatomical differences between lianas and trees. As canopy lianas have lower mechanical requirements than trees, they typically allocate more resources to wood storage or water transport, rather than to mechanical support [12]. However, our results indicate that these wood allocation differences between lianas and trees did not hold true at the seedling stage, when most liana seedlings are self-standing and, therefore, appear to be functionally similar to trees. For instance, we found non-significant differences between liana and tree seedlings in terms of the stem area allocated to fibers, fiber wall thickness, and wood density, indicating similar mechanical needs. Similarly, liana and tree seedlings allocated comparable fractions of stem area to vessels and parenchyma cells, which suggests that their water transport and storage capacities are alike. Beyond the similarity in terms of water transport capacity (i.e., vessel lumen size and fraction), liana and tree seedlings were also similar in their hydraulic safety, with both groups showing comparable intervessel pit diameters. This is contrary to previous studies on sapling and adult lianas that have reported lower hydraulic safety for lianas [30,36]. In addition to the mentioned hydraulic characteristics, at a structural level it is known that parenchyma, besides its key role in storage (e.g., [31]), is also thought to favor stem liana bending and twisting [12,13]. Possibly, lianas require larger amounts of parenchyma cells during their climbing phase, but not when they are self-supporting.

Aside from the quantitative traits we discussed above, some liana species are also characterized by qualitative anatomical features, such as the presence of cambial variants [12,37]. Cambial variants are an alternative way of secondary growth that differs from the traditional form of secondary growth found in most trees [12,37] and can favor wound healing and stem bending without compromising hydraulic functioning or long-distance photosynthate transport by increasing the relative amount of phloem (e.g., [38]). It has been suggested that some cambial variants are absent when lianas are self-standing, and instead are developed once the climbing stage starts [12]. While we observed cambial variants in a couple of liana species (i.e., interxylary phloem islands in *Pisonia aculeata*, and phloem wedges in *Abrus precatorious*; Appendix A), most of our study species lacked these features. For example, from an anatomical point of view, Bignoniaceae lianas are distinguished by the presence of four, or multiples of four, equidistant phloem wedges (e.g., [39]). However, in *Bignonia pterocalyx* seedlings, these phloem wedges were either incipient or entirely absent (See Appendix A). Similarly, lianas of the genus *Machaerium* are known for having flattened stems that result from successive cambia (e.g., [40]), but *Machaerium mycrophyllum* seedlings displayed rounded stems and no successive cambia (See Appendix A). The absence, or incipient development, of cambial variants in most of the liana seedling species reinforces the notion that the liana and tree seedlings we studied were anatomically similar. This agrees with some studies suggesting that self-supporting lianas can be morphologically and functionally similar to trees (e.g., [1,21]), but it is contrary to a recent study showing that self-supporting lianas have different leaf architectural traits in comparison to trees [8]. We suggest further studies to investigate the extent to which self-supporting liana and tree seedlings differ in other axes of functional trait variation, such as leaf and root economic spectra.

### 3.2. Demographic Differences Between Liana and Tree Seedlings

Lianas are generally thought to grow faster than trees [16], and our results support this notion. We observed that lianas displayed a growth advantage relative to trees. This faster growth may be linked to their larger variation in hydraulically weighted vessel diameters, which may help optimize water transport efficiency, particularly in seasonal environments such as tropical dry forests [41]. During the wet season, lianas can benefit from their wider vessels, which facilitate efficient water transport. In contrast, during the dry season, they may reduce the risk of embolism by utilizing their narrower vessels, allowing them to maintain water transport despite reduced water availability [26,29,30]. To further explore this, we tested the relationship between growth and variance in hydraulically weighted vessel diameters and found a positive significant effect (*t*-value: 4.642, *p*-value: 7.55 × 10^−6^), reinforcing the link between growth and water transport. Also, other unmeasured traits, such as deeper rooting systems ([16], but see [15]), may also contribute to the observed growth advantage. Together, transport traits could confer a competitive advantage to under both dry and wet conditions.

However, despite this apparent growth advantage, we did not observe differences in survival rates between lianas and trees. This disconnect between growth and survival has been reported in other studies as well [23], highlighting that the factors driving these two demographic processes are not necessarily coupled. One possible explanation is that while traits favoring rapid growth may enhance competitive ability for light and water, they do not necessarily increase survival under all environmental conditions. Survival may be influenced by other factors, such as disease, which could be more closely linked to the mechanical properties of wood [42,43]. Since we did not observe differences in fiber and parenchyma traits between lianas and trees, this could explain the similarity in survival rates between trees and lianas. At this early stage, the stems might be particularly susceptible to natural enemies, and fiber and parenchyma traits may provide physical protection and an active response against pathogens respectively [44,45,46].

It is also important to recognize that the demographic data used in this study were based on a single year, which might not represent the demographic trends in the longer term for these species. Seedlings can spend decades growing in the forest understory [47,48], and their growth and survival can fluctuate substantially across years. Longer-term monitoring will be crucial to fully understand the relationship between growth and survival in liana and tree seedlings. For instance, Umaña et al. [23] found that the growth advantage of lianas was only evident in certain years, suggesting that the benefits of their traits may be contingent on specific environmental factors or periods of resource limitation.

In conclusion, we have shown that self-standing liana and tree seedlings from tropical dry forests are, overall, similar in terms of biomechanical and water storage properties of wood as well as in their survival rates. This suggests that the well-known anatomical differences between lianas and trees may possibly depend, among other factors, on liana developmental stage. Specifically, it is likely that these differences emerge once lianas start their climbing phase. However, we also observed that lianas displayed a growth advantage potentially associated with water transport strategies. Combined, our findings suggest that trees and self-supporting liana seedlings might have overlapping niches in terms of their biomechanical support and water storage, which can have important implications for understanding the interactions between these two life forms, as well as their relative abundance, at the seedling stage.

## 4. Materials and Methods

### 4.1. Study Sites

The study was conducted in three tropical dry forests in Colombia (Table 3). These forests represent a precipitation gradient ranging from 899.4 mm per year in Tayrona, to 1528.4 mm per year in Colorados. In each one of these forests, a 1-ha permanent plot was established between 2013 and 2014 [49]. Subsequently, in 2021, 100 1-m^2^ plots were established in each 1-ha plot to study the dynamics of seedlings.

### 4.2. Species Selection

We randomly selected 10 species of lianas and 10 species of trees (Table 4, Figure 3, also see Appendix A) from the most abundant species at the seedling stage based on census data from the 1 m^2^ plots that have been established in each site. These seedling species were chosen to represent taxa widely related phylogenetically. For trait collection, we harvested two to six seedlings (≥20 cm and ≤130 in height) per species from the forests encircling each 1-ha plot. We cut a 5-cm sample from the base of each seedling stem and divided it into two halves. One of these segments was used to measure wood density, while the other was kept, until further processing in the laboratory for measuring wood anatomical traits, in a glass container filled with a 50:50 solution of 96% ethanol and water.

### 4.3. Trait Measurement

For every stem segment, the wood density was measured by dividing the fresh volume by the dry mass. Fresh volume was measured by the water displacement technique, and then the samples were oven-dried to a constant mass at 103 °C to calculate the dry mass [50]. Each stem segment was sectioned into cross and tangential anatomical sections (10–15 µm thick) using a rotary microtome equipped with disposable blades (Leica RM2255; Leica Microsystems, Wetzlar, Germany). Anatomical slices were stained with 1% Safranine for 10 min and 1% AstraBlue for 10 min. Following a one-minute ethanol series dehydration process at 50%, 75%, and 96%, they were dipped in solvent and placed on slides using Eukitt mounting media (Electron Microscopy, Hatfield, PA, USA).

Photographs of anatomical slides were taken with a camera (Axiocam 305 color, Zeiss, Jena, Germany) attached to an Axioscope A1 light microscope. One 10× and one to three 40× (depending on stem diameter) cross-sectional photos were obtained from each anatomical slide, in addition to one 100× tangential shot. To capture cross-sectional images, we carefully chose a representative section from each anatomical slide that, in general, included most of the radial variation (that is, from pith to bark) in the wood structure while omitting wounds or tension wood.

We measured 12 wood anatomical traits associated with storage, water transport, and mechanical support (three to five individuals per species) (Table 5). Using a drawing pad (Wacom CTL-472; Beijing, China) and Photoshop CS4 (Adobe Systems Incorporated, San Jose, CA, USA), we manually colored each cell type in the 10× cross-section pictures in order to calculate the fractions (i.e., percentage of stem cross-sectional area) of vessels, fibers, and parenchyma cells. Afterward, we used the batch function in the ImageJ 1.52 program https://imagej.net (accessed on 22 October 2024), to automatically compute the fractions. We examined tangential images to confirm the classification of wood cell types in cases where it was difficult to distinguish between fibers and axial parenchyma cells in the cross-sectional photos because of fiber dimorphism and/or thick-walled parenchyma [51].

Each 40× cross-sectional image was split into four equal parts in order to measure the fiber wall thickness (Fwt). Twenty fibers were randomly chosen and measured for each part, for a total of 80 fibers in each image. Intervessel pit diameter aperture was measured on three pits per vessel and three vessels per individual [52]. Mean hydraulically weighted vessel diameter (Dh) was calculated as: Dh = (∑ D4/n)1/4, where n is the total number of vessels measured and D is the average of the major and minor axes for each vessel cross-section [53,54]. We measured on average 142 vessels per photograph (range 15–455). Fwt, Dpm and Dh were measured in ImageJ software. Vessel clustering index (Vci) [26] was calculated by dividing the total number of vessels into the total number of clusters in each slice. Vessel diameter variance was calculated using the following formula: VarDh = (∑ (Di − D)^2^)/n − 1, where Di represents the diameter of each vessel, D is the arithmetic mean of all xylem vessel diameters, and n is the total number of xylem vessels measured.

### 4.4. Seedling Growth and Survival

Seedling survival was monitored during one year. Seedling height relative growth rate (RGR, cm·cm^−1^ y^−1^) for each seedling during a one-year period was estimated as follows: ln (Hf/Hi)/∆t, where Hf and Hi denote the final and beginning stem heights, respectively, and ∆t represents the interval between the height measurements [55].

### 4.5. Light Conditions

As light is one key limiting resource determining seedling growth [16,22] we measured understory light conditions at each seedling plot by taking hemispherical photographs with a fish-eye lens (Criacr, Amir Technology Co., Ltd., Dongguan, China) with a 180-degree angle of view that was adjusted for a cell phone that was set up on a tripod one meter above the forest floor. To get consistent lighting conditions, photos were taken in May and July of 2021 near the middle of each 1 m^2^ seedling plot, either at dawn or dark. These photos were then analyzed to calculate a gap light index (GLI). These images were turned into black and white, and the percentage of white pixels in each image was used to determine the GLI.

### 4.6. Data Analyses

To visualize the possible differences between liana and tree seedlings in relation to wood anatomical traits, we first performed a principal component analysis (PCA) with species trait means as data points. Then, to assess the differences between the two life forms in terms of wood anatomical traits, we first performed a normality test and then conducted a two-tailed *t*-test for each trait separately with species mean values as data points. To examine whether liana and tree seedlings had different relative growth rates, we ran a mixed-effects model predicting seedling individual RGR based on life form (i.e., liana or tree). In this mixed-effects model, we included seedling height and canopy light (i.e., GLI) as fixed effects, and species as a random effect. To analyze the difference in survival between liana and tree seedlings, we ran generalized mixed-effects models predicting seedling survival with life form, seedlings height, and GLI as fixed effects, and species as random effects. RGR, as well as Wd, Dh, Vf, Dpm, and Rpf were log-transformed to meet normality assumptions. All analyses were performed in the software R version 4.3.2 (R Development Core Team, Vienna, Austria, 2023).

## Figures and Tables

**Figure 1 plants-13-03023-f001:**
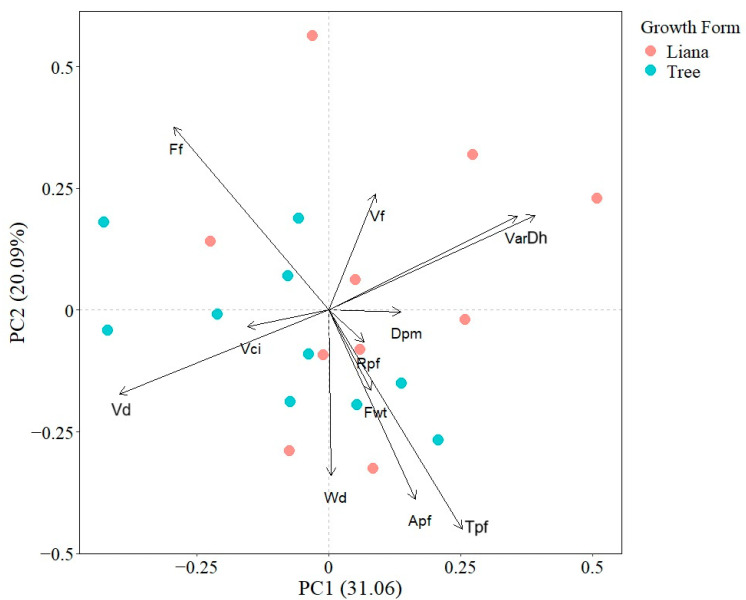
Principal component analysis showing the associations between seedling wood traits. Points represent species mean values. Apf (Axial parenchyma fraction), Dh (Mean hydraulically weighted vessel diameter), Dpm (Horizontal pit membrane diameter aperture), Ff (Fiber fraction), Fwt (Fiber wall thickness), Rpf (Radial parenchyma fraction), Tpf (Total parenchyma fraction), VarDh (Variance in vessel diameter), Vci (Vessel clustering index), Vd (Vessel density), Vf (Vessel fraction), and Wd (Wood density).

**Figure 2 plants-13-03023-f002:**
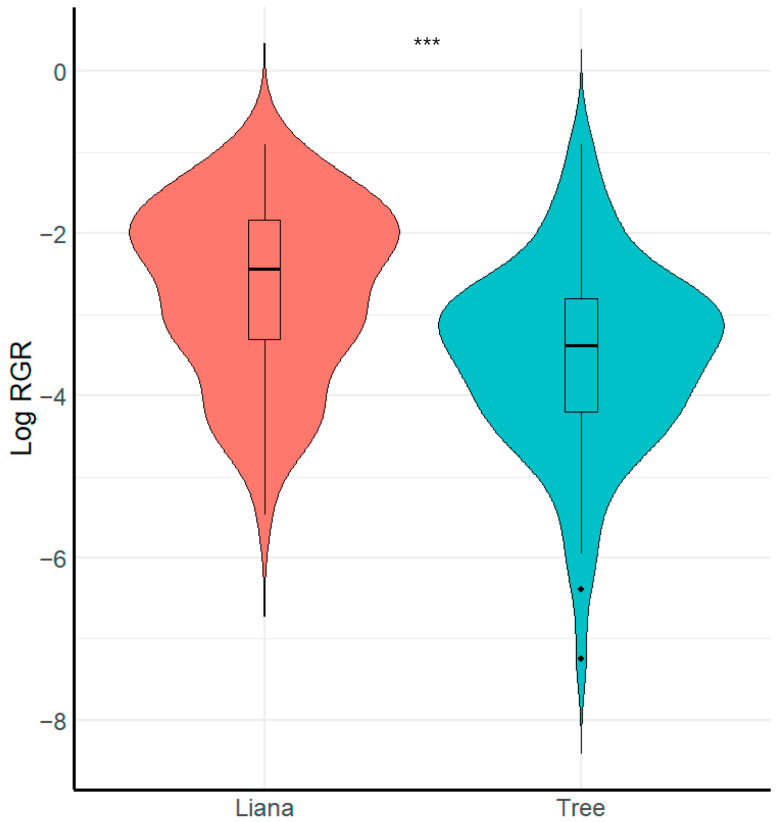
Boxplot showing comparison of seedling relative growth rate RGR between growth forms. *** = Significant differences (*p* < 0.001).

**Figure 3 plants-13-03023-f003:**
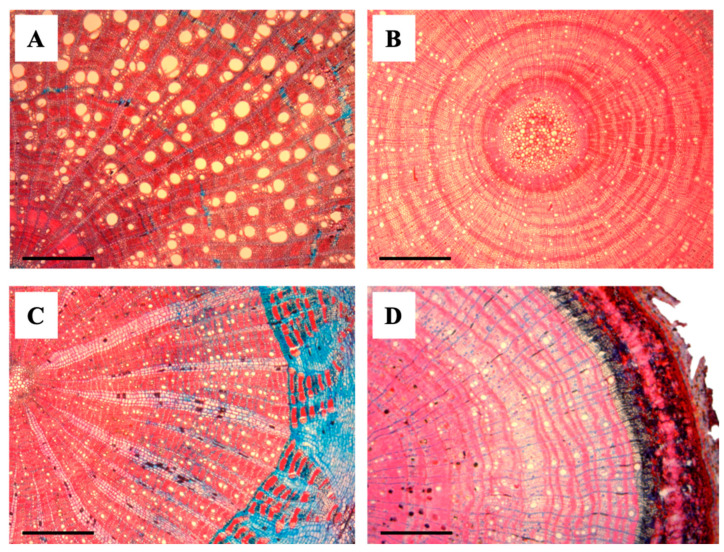
Transverse anatomical sections of liana and tree seedlings. (**A**) *Abrus precatorius* (Liana), (**B**) *Paullinia* sp. (Liana), (**C**) *Oxandra venezuelana* (Tree), and (**D**) *Trichilia acuminata* (Tree). Scale bars = 500 μm.

**Table 1 plants-13-03023-t001:** Comparison of wood anatomical traits between liana and tree seedlings using *t*-test. Bold values indicate significant (*p* < 0.05) differences.

Trait	Unit	*t*-Value	*p*-Value	Mean	Range
Mean hydraulically weighted vessel diameter	µm	1.815	0.089	47.124	24.365–121.283
Vessel fraction	%	−0.091	0.928	7.423	2.897–20.522
Vessel density	Vessels per mm^2^	−2.395	**0.028**	53.794	5.671–198.8
Horizontal pit membrane diameter aperture	µm	−0.293	0.774	7.533	5.008–11.712
Variance in vessel diameter	unitless	2.122	**0.048**	121.22	3.15–885.353
Vessel clustering index	unitless	−0.389	0.702	1.384	1–4.04
Fiber wall thickness	µm	0.397	0.696	3.798	2.573–8.56
Fiber fraction	%	−0.41703	0.6817	54.952	33.469–83.461
Wood density	g/cm^−3^	−0.274	0.788	0.563	0.281–0.665
Axial parenchyma fraction	%	−0.934	0.363	23.503	0.0652–43.238
Radial parenchyma fraction	%	1.551	0.139	15.651	4.049–45.58
Total parenchyma fraction	%	0.184	0.855	37.625	10.643–56.75

**Table 2 plants-13-03023-t002:** Summary values of relative growth rate (RGR) on seedlings of the species selected. Mean, standard deviation (Sd), and range are shown.

Growth Form	Family	Species	Mean	Sd	Range
Liana	Apocynaceae	*Forsteronia spicata*	0.125	0.103	0.004–0.408
	Bignoniaceae	*Bignonia pterocalyx*	0.088	0.051	0.009–0.152
	Celastraceae	*Hippocratea volubilis*	0.039	0	0.039–0.039
	Fabaceae	*Abrus precatorius*	-	-	-
	Fabaceae	*Machaerium isadelphum*	0.038	0.013	0.024–0.051
	Nyctaginaceae	*Pisonia aculeata*	0.045	0.043	0.009–0.108
	Sapindaceae	*Paullinia aff. cururu*	0.040	0	0.040–0.040
	Sapindaceae	*Paullinia cururu*	0.108	0.076	0.014–0.200
	Sapindaceae	*Paullinia* sp1	0.010	0.069	2.928–3.066
	Sapindaceae	*Paullinia* sp2	0.068	1.362	0.103–3.060
Tree	Annonaceae	*Oxandra venezuelana*	0.026	1.441	0.012–2.993
	Fabaceae	*Coursetia ferruginea*	0.052	0.011	0.042–0.063
	Fabaceae	*Inga edulis*	0.008	0	2.575–2.575
	Fabaceae	*Neltuma juliflora*	0.067	0	0.067–0.067
	Fabaceae	*Pterocarpus rohrii*	0.068	0.062	0.003–0.151
	Malpighiaceae	*Malpighia glabra*	0.074	0.178	0.001–0.804
	Meliaceae	*Trichilia acuminata*	0.051	1.464	0.006–3.095
	Phyllanthaceae	*Phyllanthus botryanthus*	0.012	0.003	0.009–0.015
	Rutaceae	*Amyris pinnata*	0.058	0.046	0.018–0.134
	Sapindaceae	*Melicoccus bijugatus*	0.048	0.036	0.005–0.171

**Table 3 plants-13-03023-t003:** Description of the study sites [49].

Plot	Altitude (AMSL)	Coordinates	Total Annual Precipitation (TAP; mm)	Mean Annual Temperature (MAT; °C)
Tayrona	15	11.31° N, −74.13° W	899.4	27.38
Colorados	301	9.94° N, −75.11° W	1528.4	26.1
Cotové	385	6.53° N, −75.83° W	1193.8	26.92

**Table 4 plants-13-03023-t004:** Family and plot location of the study species of lianas and trees.

Growth Form	Family	Species	Plot
Liana	Apocynaceae	*Forsteronia spicata* (Jacq.) G. Mey.	Cotové
	Bignoniaceae	*Bignonia pterocalyx* (Sprague ex Urb.) L.G. Lohmann	Tayrona
	Celastraceae	*Hippocratea volubilis* L.	Tayrona
	Fabaceae	*Abrus precatorius* L.	Cotové
	Fabaceae	*Machaerium microphyllum* (E. Mey.) Standl.	Cotové
	Nyctaginaceae	*Pisonia aculeata* L.	Cotové
	Sapindaceae	*Paullinia aff. cururu* L.	Tayrona
	Sapindaceae	*Paullinia cururu* L.	Cotové
	Sapindaceae	*Paullinia* sp1	Colorados
	Sapindaceae	*Paullinia* sp2	Colorados
Tree	Annonaceae	*Oxandra venezuelana* R.E. Fr.	Colorados
	Fabaceae	*Coursetia ferruginea* (Kunth) Lavin	Tayrona
	Fabaceae	*Inga vera* Willd	Colorados
	Fabaceae	*Prosopis juliflora* (Sw.) DC.	Tayrona
	Fabaceae	*Pterocarpus rohrii* Vahl	Tayrona
	Malpighiaceae	*Malpighia glabra* L.	Cotové
	Meliaceae	*Trichilia acuminata* (Humb. & Bonpl. ex Roem. & Schult.) C. DC.	Colorados
	Phyllanthaceae	*Phyllanthus botryanthus* Müll. Arg.	Cotové
	Rutaceae	*Amyris pinnata* Kunth	Cotové
	Sapindaceae	*Melicoccus bijugatus* Jacq.	Cotové

**Table 5 plants-13-03023-t005:** Wood anatomical traits measured on ten liana and ten tree seedling species.

Trait (Abbreviation)	Unit	Description	Function
Axial parenchyma fraction (Apf)	%	Percentage of stem cross-sectional area allocated to axial parenchyma	Storage and structural flexibility
Mean hydraulically weighted vessel diameter (Dh)	µm	Mean diameter that all of the vessels in a stem would have in order to correspond to the overall conductivity for the same numbers of conduits	Water transport efficiency and safety
Horizontal pit membrane diameter aperture (Dpm)	µm	Horizontal pit membrane diameter	Water transport safety
Fiber fraction (Ff)	%	Percentage of stem cross-sectional area allocated to fibers	Mechanical stability
Fiber wall thickness (Fwt)	µm	Double wall between adjacent fibers	Mechanical stability
Radial parenchyma fraction (Rpf)	%	Percentage of stem cross-sectional area allocated to radial parenchyma	Storage and structural flexibility
Total parenchyma fraction (Tpf)	%	Percentage of stem cross-sectional area allocated to total parenchyma	Storage and structural flexibility
Variance in vessel diameter (VarDh)	Unitless	Vessel diameter variance	Water transport efficiency and safety
Vessel clustering index (Vci)	Unitless	Total number of vessels divided by the number of vessel groups.	Water transport efficiency and safety
Vessel density (Vd)	Vessels per mm^2^	Number of conduits per cross-sectional area	Water transport safety
Vessel fraction (Vf)	%	Percentage of stem cross-sectional area allocated to vessels	Water transport capacity
Wood (density (Wd))	g/cm^−3^	Oven-dry mass divided by saturated volume of the wood section.	Mechanical stability Water transport safety

## Data Availability

The data that support the findings of this study are available at https://doi.org/10.7302/c4f7-bn13.

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
