# Peer review of "Wood Anatomical and Demographic Similarities Between Self-Standing Liana and Tree Seedlings in Tropical Dry Forests of Colombia"

_plants, 2024, doi:10.3390/plants13213023_

Round 1
Reviewer 1 Report
Comments and Suggestions for Authors
My general opinion:
I consider plant histology studies to be important and I believe that even today they can provide new, useful results. The results of the manuscript shade and clarify our generally learned/taught knowledge about the secondary thickening of the stems, and within about liana-type thickening.
„Allocate” - It is a frequently used word in the manuscript, and I think that its use is sometimes not justified, it does not facilitate the understanding of the given statement. For example, in line 74, 206.
I suggest based for example on the article of Smith K.T (Compartmentalization, Resource Allocation, and Wood Quality. Curr Forestry Rep (2015) 1:8–15. in which the “within-tree allocation” is used as “within-tree allocation between tree growth and protection from herbivory”) the use of „within-stem allocation (of resources)” ….. between/among investigated functions in lianas and trees… Since a stem fulfills several functions, and the ratio of the tissues serving these can vary.
Line 171: Important to highlight, that the anatomical features were investigated only in wood/xylem. In my opinion phloem and cortex in a young seedling can contribute significantly to transport, storage, photosynthesis, tissue replacement caused by injury, flexibility, etc.
Discussion
To lines 221-245. It is surprising that the development of the genetically determined histological structure is so dependent on the environment, in this case, whether there is support or not. It would be exciting to examine the genetic background of this as well in the future.
Line 251-257: What does „demographic difference” between lianas and tress means? And what is the connection between demographic and structure of wood? In my opinion demographic is more than ability of species to grow! Abiotic and biotic environmental factors influence it.
It is not surprising that liana seedlings do not grow quickly without support. Plants do not waste energy unnecessarily, they "wait" for help.
Growth can also be strongly influenced by light conditions. Did the light conditions also change during the dry season? The description of the places of collection/structure of the forests is missing.
4. Materials and Methods
line 306: „(≥20 cm and ≤130 in height) -in this form these data are not understandable! Height of plantlets were 20-130 cm.? In my opinion if it is true, this is a too wide range of data! - Because, the structure of stems of 20 cm or of 130 cm (belonging to same species) show lots of differences. In „Discussion” (line 197) is written: „that we compared liana and tree seedling of similar heights, and measured wood anatomical traits at the same stem height (i.e., at 5 cm from the base)”. This statement is very important, because recently, several studies brought a new perspective on the inter-individual variation of wood anatomical traits highlighting the role of tree height as the driver of wood anatomical variation along plant stem, as opposed to the classical pith-to-bark age trend.
May I ask how the two statements are consistent?
Figure 3. and Fig. S2: Beautiful pictures of very different stem and xylem structures! How old are these plantlets?
Table S1: Does the fourth coloumn contain the data of lianas and trees as well?
Table S2 – What is the guiding principle in the enumeration?
I indicated my minor corrections in the text! (manuscript v5)

Author Response
- I consider plant histology studies to be important and I believe that even today they can provide new, useful results. The results of the manuscript shade and clarify our generally learned/taught knowledge about the secondary thickening of the stems, and within about liana-type thickening. „Allocate” - It is a frequently used word in the manuscript, and I think that its use is sometimes not justified, it does not facilitate the understanding of the given statement: For example, in line 74 206. I suggest based for example on the article of Smith K.T (Compartmentalization, Resource Allocation, and Wood Quality. Curr Forestry Rep (2015) 1:8–15. in which the “within-tree allocation” is used as “within-tree allocation between tree growth and protection from herbivory”) the use of „within-stem allocation (of resources)” ….. between/among investigated functions in lianas and trees… Since a stem fulfills several functions, and the ratio of the tissues serving these can vary.
Authors: We use the word "allocation" to describe how plants balance the distribution of their biomass or limited resources—such as energy and nutrients—across various functions. We believe that this usage is appropriate within the context of our research, as it captures the dynamic nature of resource distribution in plants (trees and lianas). That said, we also recognize the validity of your suggestion. After careful consideration, we have opted to retain the term "allocation," as it can be applied broadly across different contexts, including the one you proposed.
- Line 171: Important to highlight, that the anatomical features were investigated only in wood/xylem. In my opinion phloem and cortex in a young seedling can contribute significantly to transport, storage, photosynthesis, tissue replacement caused by injury, flexibility, etc.
Authors: We indicated throughout the text that we measured anatomical traits in wood (L:175; L:178).We agree with the reviewer that bark, including phloem and cortex, can play an important role in the functioning of seedlings. However, our study is focused on wood.
Discussion
- To lines 221-245. It is surprising that the development of the genetically determined histological structure is so dependent on the environment, in this case, whether there is support or not. It would be exciting to examine the genetic background of this as well in the future.
Authors: In addition to environmental drivers, there may be additional intrinsic drivers linked to developmental shifts that occur during ontogeny. We agree that studying the underlying causes of these shifts would be interesting but this falls out of the scope of this manuscript.
- Line 251-257: What does „demographic difference” between lianas and trees means? And what is the connection between demographic and structure of wood? In my opinion demographic is more than ability of species to grow! Abiotic and biotic environmental factors influence it. It is not surprising that liana seedlings do not grow quickly without support. Plants do not waste energy unnecessarily, they “wait” for help. Growth can also be strongly influenced by light conditions. Did the light conditions also change during the dry season? The description of the places of collection/structure of the forests is missing.
Authors: In our study, when we refer to "demographic differences," we are specifically discussing differences in growth rates between lianas and trees. For clarity, we have used the term “growth” instead of “demography”. We agree that demography encompasses more than just growth; and that both biotic and abiotic factors affect growth. In the models explaining growth based on growth form (i.e., liana or tree), we included seedling height as a fixed factor and species as a random effect.
Regarding your point on light conditions and environmental influences, while our study primarily focuses on seasonal water availability, we agree that light can also play a critical role. We have measurements of light availability per 1 m2 plot that have been included as an additional fixed effects in our models. The results show that light (i.e., canopy gap index) is unrelated to growth (F=1.67; p>0.05). Moreover, including light in the models did not change considerably the marginal or conditional coefficients of determination (R2): Without light: R2 conditional: 0.41 and R2 marginal:0.09; with light: R2 conditional: 0.38 and R2 marginal: 0.10
What does „demographic difference” between lianas and trees means? And what is the connection between demographic and structure of wood?:
Authors: The term demography refers to growth. For clarity, we have used the term “growth” instead of “demography” in the discussion.
Did the light conditions also change during the dry season?:
Authors: The light conditions likely varied between the dry and wet seasons, as these forests are deciduous. However, this temporal variation in light should affect both trees and lianas similarly. We believe that spatial variation in light may play a more significant role in giving some seedlings an advantage over others. To account for this potential effect, we have introduced new models that include the percentage of understory light availability at the seedling plot level (L:147-150).
Materials and Methods
- line 306: „(≥20 cm and ≤130 in height) -in this form these data are not understandable! Height of plantlets were 20-130 cm.? In my opinion if it is true, this is a too wide range of data! - Because, the structure of stems of 20 cm or of 130 cm (belonging to same species) show lots of differences. In „Discussion” (line 197) is written: „that we compared liana and tree seedling of similar heights, and measured wood anatomical traits at the same stem height (i.e., at 5 cm from the base)”. This statement is very important, because recently, several studies brought a new perspective on the inter-individual variation of wood anatomical traits highlighting the role of tree height as the driver of wood anatomical variation along plant stem, as opposed to the classical pith-to-bark age trend. May I ask how the two statements are consistent?
Authors: We have adopted the format suggested by the reviewer. Regarding your other comment, this seedling height range is commonly used to define seedlings in permanent forest surveys (see ForestGEO protocol [1] https://forestgeo.si.edu/protocols/seedling-performance . In response to your question about standardization, we agree that wood anatomical structure can change with plant height. This is particularly true in the case of vessel lumen size, as plants need to build narrower vessels as they grow in height. To take into account these height-related variations, we standardized all our measurements by size, as all cross-sections were cut at 5 cm from the stem base. This information is included in the methods section (L:312).
- Figure 3. and Fig. S2: Beautiful pictures of very different stem and xylem structures! How old are these plantlets?
Authors: Thank you. These seedlings were collected from the field and we do not have accurate information on their age.
- Table S1: Does the fourth coloumn contain the data of lianas and trees as well?
Authors: Yes, the range of each trait was calculated with the values of both lianas and trees.
- Table S2 – What is the guiding principle in the enumeration?
Authors: Can you please be more specific about what you mean by enumeration?
References:
- Anderson-Teixeira, K. J., Davies, S. J., Bennett, A. C., Gonzalez-Akre, E. B., Muller-Landau, H. C., Joseph Wright, S., … Baltzer, J. L. CTFS-ForestGEO: a worldwide network monitoring forests in an era of global change. Global Change Biology, 2014. 21(2), 528–549.

Reviewer 2 Report
Comments and Suggestions for Authors
Plants Wood anatomical and growth similarities of lianas and tree seedlings in tropical dry forests of Colombia
· The manuscript has several draft changes and needs to be cleaned up.
TITLE. The plural must be corrected in the title. Suggestions:
1) Wood anatomical and growth similarities between liana and tree seedlings in tropical dry forests of Colombia.
2) Wood anatomical and growth similarities between seedlings of lianas and trees in tropical dry forests of Colombia.
Abstract: good text.
1. Introduction
· The introduction contains many unnecessary details to present the issue to be investigated. Much information could support the discussion and should be transferred.
2. Results
· Figure 1. The meaning of the abbreviations must be present in the figure legend. Each figure must be self-explanatory.
· Table 1. The units must be showed for the parameters, for example: vessel diameter (μm). Attention! Some values cannot be analyzed coherently without knowing the unit used in the parameter. Some values would certainly not be in the order of thousands of micrometers. Perhaps the comma is being used incorrectly in some numbers, as occurred with the Supplementary Material.
· The manuscript describes ten species of lianas. However, there are only 9 lianas in Table S2. Were all results obtained with ten or nine species? You should clarify it. Furthermore, there are doubts on the identity of Paullinia aff. cururu, Paullinia cururu, Paullinia sp1, and Paullinia sp2. The expertise of a taxonomist may help with this problem.
3. Discussion
· Good. There are several important insights in the discussion. The results are well discussed and related to the literature.
· What is the significance of these results? What effects do wider vessels promote? What are the adaptive advantages of these characteristics? An evolutionary perspective is necessary.
4. Materials and Methods
· Some undetermined species could be identified by a taxonomist (sp1, sp2, and Paullinia aff. cururu.
· Table 3. All species must be with their authorities. You should separate the table rows into two groups (lianas versus trees) and put the families in alphabetical order. Do not use italics for sp1 and sp2. The following species must be revised:
1) Prosopis juliflora (Sw.) DC. is a synonym of Neltuma juliflora (Sw.) Raf.
2) Inga vera Kunth is a nom. illeg., and this name is a synonym of Inga edulis Mart.
3) Machaerium microphyllum (E.Mey.) Standl. is a synonym of Machaerium isadelphum (E.Mey.) Amshoff
· What is the concentration of the used dyes?
· How many repetitions were used for each parameter evaluated? In other words, how many individuals were evaluated for each species?
References: some references out of the pattern of the “Plants” magazine; magazine number and pages are missing in some references; there are books without the city of the edition. All references must be checked.
SUPPORTING INFORMATION
· Figure S1. All species must be in italics.
· Table S1. The meaning of the abbreviations must be present in the table. Each table must be self-explanatory. The decimal separator is a period, and you should not use a comma in this case.
· Table S2. You should separate the table rows into two groups (lianas versus trees) and put the families in alphabetical order. Do not use italics for sp1 and sp2. The decimal separator is a period, and you should not use a comma in this case.

Minor editing of English language required. Attention: the title needs correction.
Author Response
Reviewer 2
TITLE. The plural must be corrected in the title. Suggestions:
- Wood anatomical and growth similarities between liana and tree seedlings in tropical dry forests of Colombia.
Authors: We included the corrected plural in the title
Introduction
- The introduction contains many unnecessary details to present the issue to be investigated. Much information could support the discussion and should be transferred.
Authors: Could you please provide more specific details about which sections or points you consider unnecessary? We do not see any redundant information in our introduction. This would help us better understand your perspective and make the necessary adjustments to enhance the clarity and focus of the introduction.
Results
- Figure 1. The meaning of the abbreviations must be present in the figure legend. Each figure must be self-explanatory.
Authors: The abbreviations meanings were included in Figure 1.
- Table 1. The units must be shown for the parameters, for example: vessel diameter (μm). Attention! Some values cannot be analyzed coherently without knowing the unit used in the parameter. Some values would certainly not be in the order of thousands of micrometers. Perhaps the comma is being used incorrectly in some numbers, as occurred with the Supplementary Material.
Authors: The units were added to table 1. Also, the use of the comma was corrected in the traits values.
- The manuscript describes ten species of lianas. However, there are only 9 lianas in Table S2. Were all results obtained with ten or nine species? You should clarify it. Furthermore, there are doubts on the identity of Paullinia aff. cururu, Paullinia cururu, Paullinia sp1, and Paullinia sp2. The expertise of a taxonomist may help with this problem.
Authors: Thanks for catching this error. The 10th liana species was added correctly in Table S2 (Abrus precatorius).
Regarding species identification, all identifications were conducted by an expert taxonomist with extensive experience in these systems. However, since we are working with seedlings, which are non-reproductive and often lack the morphological features required for definitive species identification, some individuals could not be fully classified to the species level. To address this, we carefully differentiated these individuals into distinct morphotypes to ensure accurate representation of species-level diversity in our analyses. We are confident that this approach, combined with the expertise of the taxonomist, ensures the reliability of our species identification.
Discussion
- What is the significance of these results? What effects do wider vessels promote? What are the adaptive advantages of these characteristics? An evolutionary perspective is necessary.
Authors: In the first paragraph (L:164, L:168) we have stated: “This suggests that, at the seedling stage, lianas and trees may exhibit similar functionality in terms of their water transport and may therefore have equivalent performance. From these results we infer that the well-known anatomical and demographic differences between lianas and trees can depend, among other factors, on liana developmental stage.”
We also stated in the conclusion (L:280, L:287): “This suggests that the well-known anatomical and growth differences between lianas and trees may possibly depend, among other factors, on liana developmental stage. Specifically, it is likely that these differences emerge once lianas start their climbing phase. Also, our findings suggest that trees and self-supporting liana seedlings might have overlapping niches, which can have important implications for understanding the interactions between these two life forms, as well as their relative abundance, at the seedling stage.”
Throughout the discussion, we addressed the functional implications of the similarities in wood anatomy between lianas and trees, emphasizing that these similarities appear to lead to comparable growth rates. This contrasts with previous studies focused on adult stages, where significant wood anatomical differences between lianas and trees have been documented.
We would like to clarify that our study takes an ecological perspective rather than an evolutionary one. Thus, incorporating an evolutionary framework may fall outside the scope of our research. We would appreciate any specific suggestions on how to include this perspective in a meaningful way, particularly in relation to your comments on the significance of our results and the adaptive advantages of the observed characteristics.
Materials and Methods
- Line 292: “in 2021, 100 1-m2 plots were established in each 1-ha plot to study the dynamics of seedlings.” Is this correct?
Authors: Yes, for each 1-ha plot we established 100 sub plots of 1m2 each one.
- Some undetermined species could be identified by a taxonomist (sp1, sp2, and Paullinia aff. cururu.
Authors: Please, see answer to question 5
- Table 3. All species must be with their authorities. You should separate the table rows into two groups (lianas versus trees) and put the families in alphabetical order. Do not use italics for sp1 and sp2. The following species must be revised:
1) Prosopis juliflora (Sw.) DC. is a synonym of Neltuma juliflora (Sw.) Raf.
2) Inga vera Kunth is a nom. illeg., and this name is a synonym of Inga edulis Mart.
3) Machaerium microphyllum (E.Mey.) Standl. is a synonym of Machaerium
isadelphum (E.Mey.) Amshoff
Authors: The table was separated into lianas and trees, the families were organized in alphabetical order and the use of italics was corrected. Regarding species identification, please see answer to question 5
- What is the concentration of the used dyes?
Authors: To stain the tissues we used 1% Astra Blue and 1% Safranine.
- How many repetitions were used for each parameter evaluated? In other words, how many individuals were evaluated for each species?
Authors: We sampled between 3 - 5 individuals per species (line 349).
References
- some references out of the pattern of the “Plants” magazine; magazine number and pages are missing in some references; there are books without the city of the edition. All references must be checked.
Authors: The references were corrected according to the pattern of the “Plants” magazine.
SUPPORTING INFORMATION
- Figure S1. All species must be in italics.
Authors: The use of italics was corrected in Figure S1.
- Table S1. The meaning of the abbreviations must be present in the table. Each table must be self-explanatory. The decimal separator is a period, and you should not use a comma in this case.
Authors: The abbreviations were included in Table S1 and the commas were replaced for the period as the decimal separator.
- Table S2. You should separate the table rows into two groups (lianas versus trees) and put the families in alphabetical order. Do not use italics for sp1 and sp2. The decimal separator is a period, and you should not use a comma in this case.
Authors: The italics were removed for sp1 and sp2, and the commas were replaced for the period as the decimal separator.Also, the table rows were separated into two groups (lianas versus trees).

Round 2
Reviewer 2 Report
Comments and Suggestions for Authors
Plants Wood anatomical and growth similarities of lianas and tree seedlings in tropical dry forests of Colombia
The corrections were done, except the following.
2. Results
· Figure 1. The meaning of the abbreviations has been added. However, as noted earlier, the lists should be in alphabetical order for easy consultation.
4. Materials and Methods
· Wrong: Axio Scope; correct: Axioscope.
· Table 3. Several changes done. However, the following species must be revised. As previously pointed out:
1) Prosopis juliflora (Sw.) DC. is a synonym of Neltuma juliflora (Sw.) Raf.
2) Inga vera Kunth is a nom. illeg., and this name is a synonym of Inga edulis Mart.
3) Machaerium microphyllum (E.Mey.) Standl. is a synonym of Machaerium isadelphum (E.Mey.) Amshoff
The valid names can be checked in: https://powo.science.kew.org/ These corrections must be done in the entire manuscript.
As previously pointed out, the lists should be in alphabetical order for easy consultation. Some of them have been corrected. However, other lists have not, such as in Figure 1 and Table S1.
References: All references must be checked. See author guidelines. Also, the italics still is missing in reference 35. Machaerium multifoliolatum.
SUPPORTING INFORMATION
· Figure S1 shows a chaotic order of species. The order should be the same as in Table S2.
· As previously pointed out, the lists should be in alphabetical order for easy consultation. Some of them have been corrected. However, other lists have not, such as in Figure 1 and Table S1.

Minor editing of English language required.
Author Response
Results
- Figure 1. The meaning of the abbreviations has been added. However, as noted earlier, the lists should be in alphabetical order for easy consultation.
Authors: The meaning of the abbreviations was ordered alphabetically in figure 1 description
Materials and Methods
- Wrong: Axio Scope; correct: Axioscope
Authors: the word Axioscope was corrected
- Table 3. Several changes done. However, the following species must be revised.
1) Prosopis juliflora (Sw.) DC. is a synonym of Neltuma juliflora (Sw.) Raf.
2) Inga vera Kunth is a nom. illeg., and this name is a synonym of Inga edulis Mart.
3) Machaerium microphyllum (E.Mey.) Standl. is a synonym of Machaerium isadelphum (E.Mey.) Amshoff
The valid names can be checked in: https://powo.science.kew.org/ These corrections must be done in the entire manuscript.
Authors: The corrections of the mentioned species were done in the entire manuscript, supporting information and metadata.
- As previously pointed out, the lists should be in alphabetical order for easy consultation. Some of them have been corrected. However, other lists have not, such as in Figure 1 and Table S1.
Authors: The alphabetical order of the lists in Figure 1, Table 5 and Table S1 was corrected.
References:
- All references must be checked. See author guidelines. Also, the italics still is missing in reference 35. Machaerium multifoliolatum.
Authors: the use of italics in reference 35 was corrected
SUPPORTING INFORMATION
- Figure S1 shows a chaotic order of species. The order should be the same as in Table S2.
Authors: Figure S1 was corrected, ordered according to Table S2. Table S2 was placed in the main text due do changes in the results of the RGR analysis.
- As previously pointed out, the lists should be in alphabetical order for easy consultation. Some of them have been corrected. However, other lists have not, such as in Figure 1 and Table S1.
Authors: The lists mentioned, Figure 1, Table S1 and Table 5 were ordered alphabetically.